# Effect of Cellulose Nanocrystal Addition on the Physicochemical Properties of Hydroxypropyl Guar-Based Intelligent Films

**DOI:** 10.3390/membranes11040242

**Published:** 2021-03-29

**Authors:** Yahui Meng, Yunfeng Cao, Kaifeng Xiong, Li Ma, Wenyuan Zhu, Zhu Long, Cuihua Dong

**Affiliations:** 1Key Laboratory of Biobased Material and Green Papermaking, Qilu University of Technology, Jinan 250353, China; 18765382707@163.com (Y.M.); mlz.1219@163.com (L.M.); 2Key Laboratory of Pulp and Paper Science and Technology, Nanjing Forestry University, Nanjing 210037, China; yunfcao@163.com (Y.C.); klpp@njfu.edu.cn (W.Z.); 3School of Light Industry and Food Engineering, Guangxi University, Nanning 530004, China; xkf@qlu.edu.cn; 4Key Laboratory of Eco-Textiles, Ministry of Education, Jiangnan University, Wuxi 214122, China

**Keywords:** pH-sensing film, cellulose nanocrystal, barrier properties, stability

## Abstract

As an important functional material in food industry, intelligent packaging films can bring great convenience for consumers in the field of food preservation and freshness detection. Herein, we fabricated pH-sensing films employing hydroxypropyl guar (HPG), 1-butyl-3-methylimidazolium chloride (BmimCl), and anthocyanin (Anth). Besides, the effects of adding cellulose nanocrystals (CNC) into the composite films upon the films’ structures and physicochemical properties are elucidated. The addition of CNC promoted more compact film structures. Moreover, CNC dramatically improved several properties of the pH-sensing films, including the distinguishability of their color changes, sensitivity to pH, permeability to oxygen and water vapor, solvent resistance, durability, and low-temperature resistance. These results expand the application range of pH-sensing films containing CNC in the fields of food freshness detection and intelligent packaging.

## 1. Introduction

Effective detection of food safety issues has become increasingly important to society due to a greater number of food safety problems. Changes in pH are an important alarm that signal spoilage in many food products [1]. As an intelligent testing system of food safety, the pH indicator has been widely studied, due to its advantages of being convenient, cost-effective. However, there are some deficiencies in sensitivity. Generally, visual pH indicators consist of a pH-sensitive dye and a solid matrix that immobilizes pH dyes [2,3].

As a natural dye, anthocyanin (Anth) has been extensively used to indicate pH, due to its multiple color changes under different pH conditions: in acidic conditions, anthocyanins are mainly flavonoid cation structures; in weakly acidic conditions, the flavylium cation turns into colorless carbinol pseudobase or chalcone; and in alkaline conditions, it has a quinoidal base structure. Recently, Anth extracted from grape skins, purple sweet potatoes, black bean seed coats, and mulberries have been successfully applied in the preparation of pH-indicating smart packaging films for detecting meat freshness [4,5,6,7].

To immobilize natural pH dyes, a number of natural and biodegradable polymers, including starch [3], chitosan [8], guar gum [9], and cellulose [10], have been studied. Among these biopolymeric materials, polysaccharides have been considered among the most promising, because they are abundant, biodegradable, inexpensive and exhibit good film-forming ability. Hydroxypropyl guar (HPG) is a water-soluble polysaccharide that is suitable for biodegradable film fabrication due to its low cost, renewability, and excellent film-forming ability [11]. However, HPG films exhibit relatively poor mechanical properties and obstruct the color rendering of anthocyanin [12,13]. Great improvements in the physicochemical properties of HPG-based films are realized by combining them with other biopolymers. Cellulose nanocrystals (CNC; also called cellulose whiskers) are cellulose derivatives that are attracting considerable attention due to their many desirable properties, including renewability, biodegradability, good biocompatibility, high strength, and high optical transparency [14,15,16]. CNC films exhibit excellent mechanical strength, optical transparency, and oxygen barrier properties. They can also be used as a reinforcing agent for biopolymeric materials, providing improved physicochemical properties. Although CNC as a filler to improve the mechanical strength of HPG-based film has been studied, little has been reported in the literature about the sensitivity, temperature resistance, and light resistance of HPG-based pH-sensing films with the addition of CNC.

Herein, we report the fabrication of pH-sensing films based on HPG and CNC. 1-butyl-3-methylimidazolium chloride (BmimCl) was used as a plasticizer and tuner, and Anth was used as a pH-sensitive dye. The effects of added CNC upon the structure and physicochemical properties of the pH-sensing films were investigated. The film structures were studied by Fourier-transform infrared spectroscopy (FTIR) and scanning electron microscopy (SEM). We tested the mechanical properties and barrier properties of the pH-sensing films and examined their sensitivity and their color changes when using buffers at different pH values. We also studied the solvent resistance, temperature resistance, light resistance, and durability of the pH-sensing films.

## 2. Experimental

### 2.1. Materials

HPG with a degree of substitution of 0.15 was supplied by Wuxi JinXin Group Co. Ltd., China. The CNC suspension with a length of 200 nm, a width of 7 nm, pH of 7 (sulfuric acid hydrolysis), and a concentration of 2 wt% was purchased from Tianjin Woodelf Biotechnology Co. Ltd., China. BmimCl with a 99.0% purity was obtained from Shanghai Chengjie Chemical Co. Ltd., China. Anth (blueberry extract) with a concentration of 25% was supplied by Shandong Shengjiade Biological Co. Ltd. Buffer solutions were purchased from Shanghai Aladdin Biochemical Technology Co. Ltd., China. Ethyl alcohol, acetone, *N,N*-dimethylformamide, and dimethyl sulfoxide were obtained from Shanghai Sinopharm Chemical Reagent Co., Ltd., China. All other reagents were of analytical grade and used without any further purification.

### 2.2. Preparation of pH-Sensing Films

To produce HPG/IL/Anth film-forming solutions, 1.0 g of HPG was dissolved in 100 mL of deionized water at room temperature by stirring at 400 rpm for 5 h. After 0.1 g of BmimCl was added, the resulting solution was stirred for 1 h. Then, 20 g of Anth solution (0.005 g/mL) were added, and the resulting solution was stirred for 1 h.

To produce HPG/CNC/IL/Anth film-forming solutions, 0.6 g of HPG was dissolved in 60 mL of deionized water at room temperature by stirring at 400 rpm for 5 h. Twenty grams of a CNC suspension (2 wt%) were added, and the obtained mixture was stirred for 3 h. Then, 0.1 g of BmimCl was added, and the resulting solution was stirred for 1 h. Finally, 20 g of an Anth solution (0.005 g/mL) were added, followed by 1 h of stirring.

To produce CNC/IL/Anth film-forming solutions, 0.1 g of BmimCl was added to 50 g of CNC suspension (2 wt%), and the resulting solution was stirred for 1 h. Then, 20 g of an Anth solution (0.005 g/mL) was added, followed by 1 h of stirring.

The pH-sensing films were prepared by solution casting. Each solution was coated onto a polytetrafluoroethylene petri dish with a diameter of 150 mm and vacuum-dried at 35 °C for 2 days. The resulting films were denoted as the HPG/IL/Anth, HPG/CNC/IL/Anth, and CNC/IL/Anth films, respectively. All the films were maintained at 23 °C and a relative humidity (RH) of 63% ± 1% for 48 h before characterization. The thicknesses of the HPG/IL/Anth, HPG/CNC/IL/Anth, and CNC/IL/Anth films were 50.2 ± 1.2 μm, 50.7 ± 0.9 μm, and 49.0 ± 1.7 μm, respectively.

### 2.3. Characterization

#### 2.3.1. Thickness Measurement

The film thicknesses were determined as an average of 10 randomly selected positions on each sample using a thickness gauge (MP0, Fischer, Waldachtal, Germany).

#### 2.3.2. FTIR

The FTIR spectra for the film were recorded using a Nicolet iS 10 spectrometer with an attenuated total reflection accessory (Thermo Fisher, Waltham, MA, USA) at a resolution of 4 cm^−1^ over the 4000–500 cm^−1^ range. An average of 16 scans was reported.

#### 2.3.3. SEM

The films were affixed to a vertical brass specimen holder for cross-sectional and surface SEM imaging. The micrographs of the film samples were obtained using a Quanta 200 scanning electron microscope (Hitachi SU1510, Japan) with an acceleration voltage of 5 kV. The samples were sputtered with a thin gold coating layer prior to observation (Model ACS-4000-C4, Ulvac, Tokyo, Japan).

#### 2.3.4. Stress–Strain Measurements

The composite films were cut into rectangular strips of 15 mm × 100 mm for stress–strain tests, which were conducted at 25 °C and an RH of 30% using a BZ2.5/TNIS Zwick Material Tester (Zwick, Germany). The initial grip separation was set at 50 mm, and the specimens were loaded at a constant cross-head speed of 2 mm/min. At least 5 specimens of each sample were tested, and the average values were reported.

#### 2.3.5. Oxygen Permeability (OP)

The OPs of the films was measured with a Labthink VAC-V2 apparatus (Labthink, China) at 23 °C and an RH of 40% ± 1%. The sample area was 38.48 cm^2^, and the partial pressure of oxygen was 0.5 MPa. Three measurements were carried out for each sample, and the average values were reported.

#### 2.3.6. Water Vapor Permeability (WVP)

The WVPs of the films was measured with a Labthink W3/060 apparatus (Labthink, China) at 38 °C and an RH of 88% ± 1%. Three measurements were carried out for each sample, and the average values were reported.

### 2.4. Measurements of the Color Responses of the pH-Sensing Films in Buffers at Different pH Values

The pH-sensing films were cut into squares of 2 cm × 2 cm and submerged in buffer solutions with various pH values, i.e., 2.0, 4.0, 6.0, 8.0, 10.0, and 12.0. An image of each discolored film was obtained using an RX100 III camera. The colors of the films were determined using a portable colorimeter (Xrite2600d, MI, 101, USA) in terms of L* (lightness), a* (redness–greenness), and b* (yellowness–blueness).

### 2.5. Measurements of the Solvent Resistances and Weight Losses of the pH-Sensing Films

The pH-sensing films were cut into squares of 2 cm × 2 cm. Then, the cut film squares were immersed in ethyl alcohol, acetone, *N*,*N*-Dimethylformamide, and dimethyl sulfoxide for 48 h at 25 °C, respectively. After 48 h, they were photographed to show their solvent resistance performances. The solvent absorption ratios (%) of the films were determined by immersing the films in solvents, and they were weighed. The solvent absorption ratio (%) was calculated as Equation (1):(1)$$\mathrm{Solvent}\;\mathrm{absorption}\;\mathrm{ratio}\;=\;\frac{w_{t}{\;-\;w}_{i}}{w_{i}}\;\times \;100\%$$
where w_i_ is the initial weight of the film and w_t_ is the dried weight of the film after it is removed from the solvent.

### 2.6. Measurements of High- and Low-Temperature Resistances

The pH-sensing films were cut into squares of 2 cm × 2 cm. Then, the cut film squares were stored at −20, −10, 2, 10, 50, 70, or 90 °C for 48 h separately. The measurement of the color response was conducted as mentioned in Section 2.4.

### 2.7. Light Resistance Measurements

The pH-sensing films were cut into squares of 2 cm × 2 cm, and then, squares of the cut film were stored under room lights (25 °C; RH: 60%). Each sample was photographed and colorimetrically analyzed once a day. Measurements of the color response were conducted as described in Section 2.4.

### 2.8. Durability Measurements

The pH-sensing films were cut into squares of 2 cm × 2 cm and stored in the dark (25 °C; RH: 60%). Each sample was photographed and colorimetrically analyzed at seven-day intervals. Measurements of the color response were conducted as described in Section 2.4.

### 2.9. Measurement of Meat Freshness

Fresh chicken breast with an average weight of 60 g was purchased from the supermarket and placed in a Petri dish covered with a plastic wrap. The pH-sensing films were cut into a square of 2 cm × 2 cm and attached to the inside surface of a plastic wrap above the shrimp. The package was stored at 25 °C. The pH-sensing films on the package were observed, photographed and colorimetrically analyzed on every second day.

### 2.10. Detection of the Total Volatile Basic Nitrogen (TVBN) Levels

According to the Chinese Standard GB 5009.228-2016, the TVBN levels in chicken breast was determined.

## 3. Results and Discussion

### 3.1. Mechanical Properties and Morphologies of the pH-Sensing Films

Good mechanical properties are among the basic requirements for the films to be used as packaging. The stress–strain curves of the films are presented in Figure 1a. The tensile stress and strain were measured at break. The tensile strength, elongation at break, and Young’s modulus are shown in Figure 1b. The HPG/IL/Anth film exhibited a tensile strength of 26.41 MPa, the Young’s modulus of 1.00 GPa, and an elongation at break of 10.38%, whereas the CNC/IL/Anth film exhibited a tensile strength of 70.17 MPa, the Young’s modulus of 3.72 GPa, and an elongation at break of 3.20%. The high tensile strength and low elongation of the CNC/IL/Anth film relative to the HPG/IL/Anth film may be attributed to the high crystallinity, large aspect ratio, uniform distribution, and rigidity of CNC [17,18,19,20]. Moreover, the HPG/CNC/IL/Anth film showed a tensile stress of 45.76 MPa, the Young’s modulus of 1.84 GPa, and an elongation at break of 9.59%. This suggests that the HPG/CNC/IL/Anth film could also be a packaging film due to its high strength and good flexibility.

The cross-sectional and surface morphologies of the composite films are shown in Figure 1c. Compared with the HPG/IL/Anth film, the HPG/CNC/IL/Anth film exhibited a more uniform structure and homogeneous surface owing to the hydrogen bonding between IL, Anth, CNC, and HPG [6]. Similarly, the CNC/IL/Anth film had a smoother surface and a more uniform, dense structure compared with the HPG/CNC/IL/Anth film, which may be attributed to the uniform and dense structure of the CNC film [20]. The dense structure of the film is beneficial for obtaining good gas barrier properties.

### 3.2. Barrier Properties of the pH-Sensing Films

The two important parameters of packaging films are oxygen and water vapor permeabilities. The oxygen and water vapor permeabilities of the films are shown in Figure 2a. The HPG/IL/Anth film exhibited an oxygen permeability of 6.89 cm^3^/m^2^·day·Pa, whereas the CNC/IL/Anth film exhibited an oxygen permeability of 4.35 cm^3^/m^2^·day·Pa. The low oxygen permeability of the CNC/IL/Anth film relative to the HPG/IL/Anth film may be attributed to the dense structure of the film, the high degree of crystallinity of CNC, and the inherent impermeability of CNC, which contained tortuous pathways that impeded the spread of oxygen molecules throughout the composite film [21].

Compared with that of the HPG/IL/Anth film (719.21 g/m^2^·day), the water vapor permeability of the CNC/IL/Anth film (853.67 g/m^2^·day) was markedly higher (Figure 2b). The difference may be attributed to the fact that guar gum can effectively trap and retain the water molecules within a matrix [22], and the guar gum film shows pronounced desorption hysteresis in equilibrium moisture content and can act as barrier to air and moisture [23]. Thus, the film containing CNC achieved remarkable improvement in these barrier properties.

### 3.3. Color Response Analysis of the pH-Sensing Films to Buffers at different pH values

The color response images of the pH-sensing films are presented in Figure 3. Initially, the colors of the HPG/IL/Anth, HPG/CNC/IL/Anth and CNC/IL/Anth films were tawny, tan, and light purple, respectively. Compared with the CNC/IL/Anth film, the HPG/IL/Anth film initially exhibited a darker color due to the mixture of colors from HPG (light yellow) and anthocyanin (purple red).

As the pH value increased from 2.0 to 12.0, the HPG/IL/Anth film changed in color from light red to light brown to light olive, the HPG/CNC/IL/Anth film changed from red to light purple to olive and the CNC/IL/Anth film changed from dark red to purple to green. Compared with the HPG/IL/Anth film, the CNC/IL/Anth film exhibited easily identifiable color changes.

The color parameters L*, a*, and b* for the HPG/IL/Anth, HPG/CNC/IL/Anth, and CNC/IL/Anth films were measured (Appendix A). The HPG/IL/Anth, HPG/CNC/IL/Anth, and CNC/IL/Anth films showed high L* values, indicating that all three films exhibited high brightness. For the HPG/IL/Anth film, over the pH range from 2.0 to 12.0, the value of a* decreased from 38.41 to −6.08, while the value of b* increased from 13.83 to 26.31, indicating that the red and yellow colors diminished in intensity with increasing pH. For the CNC/IL/Anth film, over the same pH range, the value of a* decreased from 47.27 to −12.30, while b* first decreased from 20.96 to −4.28 and then increased from 1.33 to 16.14, indicating that the green color grew in intensity with increasing pH while the yellow color first weakened and then strengthened.

The color response times of the pH-sensing films are presented in Figure 4 and were 8, 6, and 2 s for the HPG/IL/Anth, HPG/CNC/IL/Anth, and CNC/IL/Anth films, respectively. Compared with the HPG/IL/Anth film, the CNC/IL/Anth film showed a faster response time. The difference may be attributed to the superior wettability of the CNC/IL/Anth film, allowing the fast absorption of the pH buffer and therefore faster color changes. Thus, the fast response and easily distinguishable color changes of the pH-sensing films presented potential for the rapid and accurate monitoring of food freshness.

### 3.4. Solvent Resistances and Weight Losses of the pH-Sensing Films

The solvent resistances of the HPG/IL/Anth, HPG/CNC/IL/Anth, and CNC/IL/Anth films were evaluated by separately immersing each film in four organic solvents, i.e., ethyl alcohol, acetone, DMF, and DMSO. The photographs of the films before and after 48 h of soaking are shown in Figure 5a. No visible changes in the structure or color of the films were observed, indicating that the HPG/IL/Anth, HPG/CNC/IL/Anth, and CNC/IL/Anth films exhibited good chemical stability and excellent solvent resistance in organic solvents.

The high solvent resistances of the films may be attributed to two reasons: (1) the good solvent resistance of guar gum against alcohols, esters, acetone, *N*,*N*-dimethylformamide, and dimethyl sulfoxide [24,25], and (2) the partial hornification of nanofibrils during the films’ drying process, as well as the films’ compact structures [26].

In addition, the solvent resistance can be quantified by measuring the change in weight of the polymer film before and after solvent immersion. The solvent absorption ratios of the HPG/IL/Anth, HPG/CNC/IL/Anth, and CNC/IL/Anth films in four organic solvents are shown in Figure 5b. The qualities of the films were increased slightly that were immersed in solvents, indicating that the films exhibited good solvent resistance.

The HPG/IL/Anth, HPG/CNC/IL/Anth, and CNC/IL/Anth films were taken out after 48 h of immersion in organic solvents and dried. After placement in a buffer at pH 12.0, the color responses of the films are shown in Figure 5c. The HPG/IL/CNC, HPG/CNC/IL/Anth, and CNC/IL/Anth films were light olive, olive, and green, respectively. Compared with the initial films, the films exhibited no visible color change after immersion, and the films still showed good responses to the indicators. These observations indicated that the organic solvents had no effect on the indicator functionality of the films and that the films exhibited excellent resistance to organic solvents. The excellent solvent resistance extends these films’ potential applications in the fields of intelligent packaging and food monitoring.

### 3.5. Temperature Resistance, Light Resistance, and Durability of the pH-Sensing Films

The HPG/IL/Anth, HPG/CNC/IL/Anth, and CNC/IL/Anth films were stored at −20, −10, 2, 10, 50, 70, or 90 °C for 48 h, and then, their temperature resistances were investigated by analyzing their color responses in a buffer at pH 12.0 (Figure 6a and Appendix A). After 48 h of storage at −20, −10, 2, or 10 °C, the colors of the HPG/IL/Anth, HPG/CNC/IL/Anth, and CNC/IL/Anth films were light olive, olive, and green, respectively. These results were consistent with the colors of the films that were stored at room temperature.

On the other hand, the colors of the pH-sensing films changed significantly with increasing temperature. After 48 h at 90 °C, the colors of the HPG/IL/Anth, HPG/CNC/IL/Anth, and CNC/IL/Anth films changed to very light olive, light olive, and pale green, respectively. These color changes may be attributed to the partial decomposition of Anth at high temperature [27]. In sum, the results indicated that the pH-sensing films exhibited good low-temperature resistance and poor high-temperature resistance.

The HPG/IL/Anth, HPG/CNC/IL/Anth, and CNC/IL/Anth films were stored at room temperature in the presence of light, and their color parameters L*, a*, and b* were measured every day (Figure 6b and Appendix A). With increasing light exposure time, the films gradually became lighter in color due to the decomposition of Anth. In addition, the films exhibited no response to the indicators after six days of light exposure. While light is essential for the biosynthesis of anthocyanin, it also accelerates its degradation. Anthocyanin retains its color significantly better when it is stored in the dark, whereas the anthocyanin content is greatly reduced when stored in light [28].

In addition, the HPG/IL/Anth, HPG/CNC/IL/Anth, and CNC/IL/Anth films were placed in long-term storage under dark conditions (25 °C: RH: 60%), and their color parameters L*, a*, and b* were measured every two weeks (Appendix A). For the CNC/IL/Anth film, following six months of storage, the values of L*, a*, and b* had scarcely changed. On the other hand, for the HPG/IL/Anth film, the values of a* and b* began to change slightly after five months of storage due to the slight decomposition of Anth.

### 3.6. Application of the pH-Sensing Films for Monitoring Chicken Breast Freshness

The freshness of chicken breast can be determined by detecting the levels of TVBN during storage. The pH-sensing films were used for monitoring the freshness of chicken breast during storage. The chicken breast showed a significant increase in TVBN levels (mg/100 g) during the storage shown as following: 6.5 (fresh) increased to 17.4 (spoiled) to 69 and then to 125 [29]. The CNC/IL/Anth film displayed a significant color change during the storage as following: light purple (0 day) to modena (1 day) and then to green (5 days) (Figure 7). The color of the CNC/HPG/IL/Anth film changed from tan to pink and then to greyish-green. The color could be easy to distinguish with the naked eye. However, the HPG/IL/Anth film showed no obvious color change. Therefore, pH-sensing films containing CNC could be applied as indicators for freshness monitoring of meat products.

## 4. Conclusions

We fabricated pH-sensing films consisting of HPG, BmimCl, and Anth. CNC was added to the films as a reinforcing agent, which can remarkably improve some properties of the pH-sensing films. For instance, the HPG/CNC/IL/Anth and CNC/IL/Anth films exhibited dense and compact structures. They also showed excellent performance for pH detection applications, including easy identification, high sensitivity, and fast color responses (6 and 2 s) to pH changes. Moreover, the films exhibited good mechanical properties, good oxygen and water vapor barrier properties, excellent solvent resistance properties, high durability (>6 months), and low-temperature (<50 °C) resistance. These results expand the application range of pH-sensing films containing CNC in the fields of food freshness detection and intelligent packaging.

## Figures and Tables

**Figure 1 membranes-11-00242-f001:**
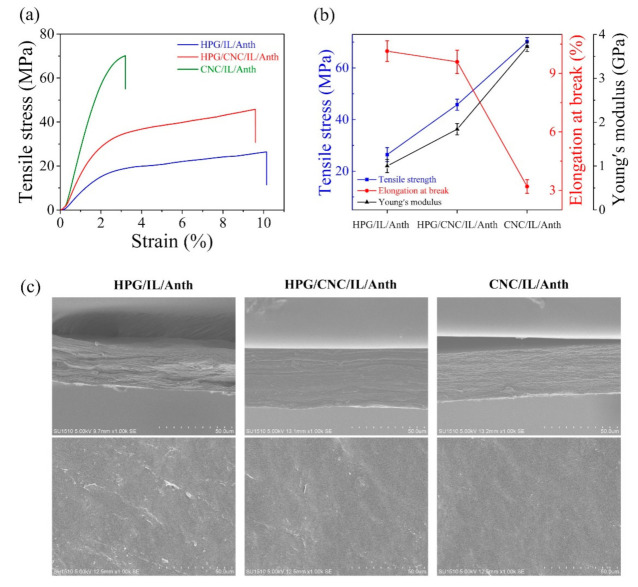
Mechanical properties and morphologies of the pH-sensing films. (**a**) Stress–strain curves of the films. (**b**) Tensile strength, Young’s modulus, and elongation at break of the films. (**c**) Cross-sectional (top) and surface (bottom) morphologies of the films.

**Figure 2 membranes-11-00242-f002:**
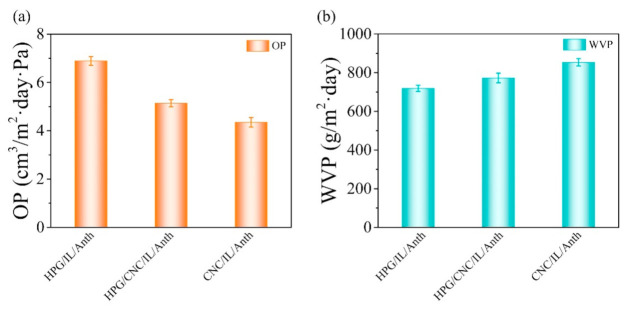
Barrier properties of the pH-sensing films. (**a**) Oxygen permeabilities of the films. (**b**) Water vapor permeabilities of the films.

**Figure 3 membranes-11-00242-f003:**
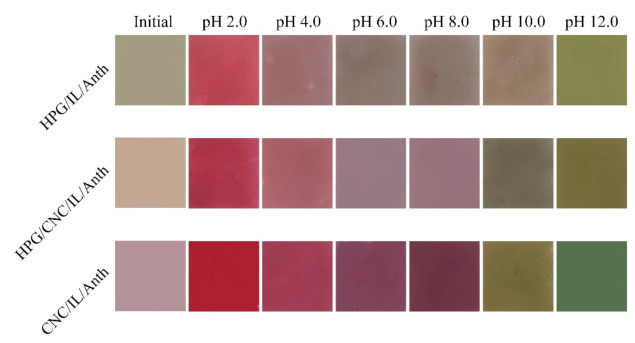
Color response photographs of the pH-sensing films in buffers at different pH values.

**Figure 4 membranes-11-00242-f004:**
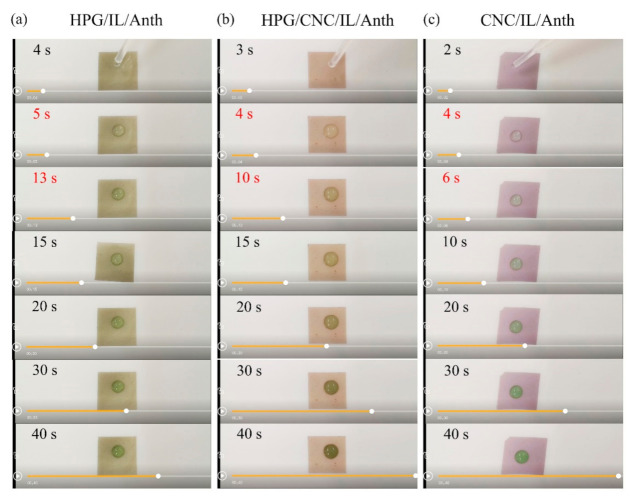
Response time of the pH-sensing films at pH 12 in buffers: (**a**) hydroxypropyl guar (HPG)/IL/anthocyanin (Anth); (**b**) HPG/cellulose nanocrystals (CNC)/IL/Anth; and (**c**) CNC/IL/Anth.

**Figure 5 membranes-11-00242-f005:**
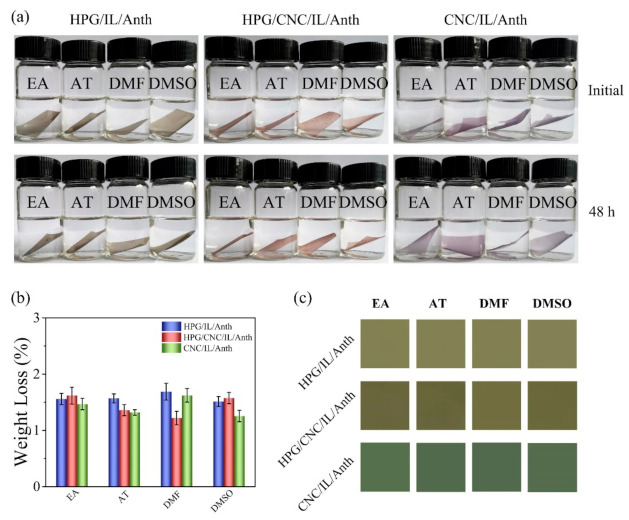
Solvent resistances of pH-sensing films. (**a**) Photographs of the films after 48 h in solvents. (**b**) The weight losses of the films after 48 h solvent treatment. (**c**) The color responses of the films in a buffer at pH 12.0 after solvent treatment. EA: ethyl alcohol; AT: acetone; DMF: *N*,*N*-dimethylformamide; DMSO: dimethyl sulfoxide.

**Figure 6 membranes-11-00242-f006:**
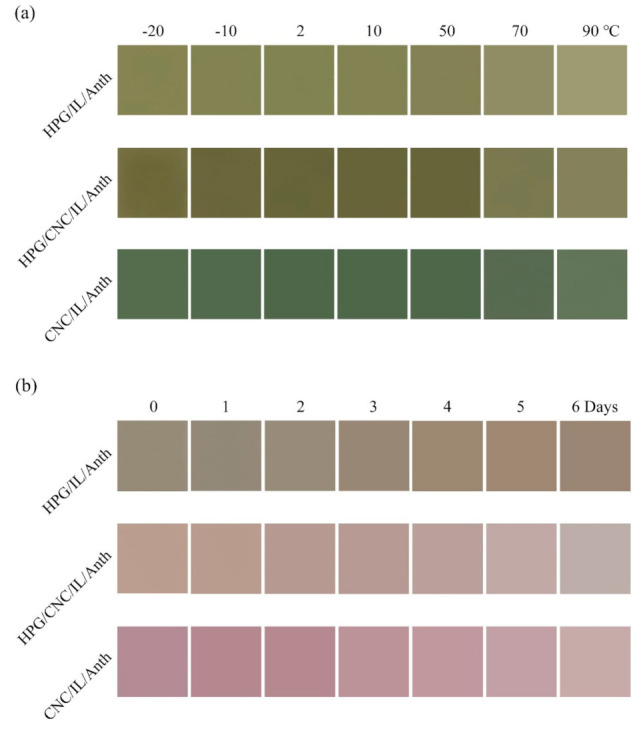
(**a**) Color response photographs of the pH-sensing films in a buffer at pH 12 and at different storage temperatures. (**b**) Color response photographs of the pH-sensing films after different light exposure times.

**Figure 7 membranes-11-00242-f007:**
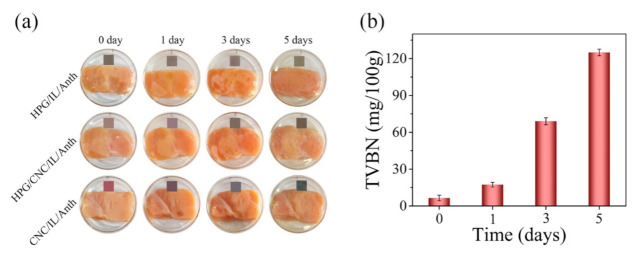
(**a**) Monitoring of the chicken breast freshness by the films. (**b**) The total volatile basic nitrogen (TVBN) levels in chicken breast stored at 25 °C for different days.

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
