# Peer review of "Effect of Cellulose Nanocrystal Addition on the Physicochemical Properties of Hydroxypropyl Guar-Based Intelligent Films"

_membranes, 2021, doi:10.3390/membranes11040242_

Round 1

Reviewer 1 Report

This manuscript described the effect of cellulose nanocrystals (CNC) on the physicochemical properties of hydroxypropyl guar-based intelligent films. In general, the experiment was well designed and the data appear to be attractive, especially the intelligent packaging has been a popular area of study. However, there are some concerns in the manuscript that need to be revised before acceptance for publication.

  1. The originality of the paper needs to be further clarified in the introduction part. Some sentences have to be added to highlight the importance of CNC and to justify the novelty of this work. For example, is there any previous work addressing the HPG-based films with CNC as nanofillers?
  2. Line 137, "where wi was the initial weight of the film and wt was the weight of the film removed from the solvent." Please clarify whether the film was dried or not after it was removed from the solvent.
  3. Figure 4a, there is not much to see in the images because the films are seen from the side, you may load that image in the supporting material if so desired, but it does not contribute to the manuscript.
  4. You are measuring swelling (or absorption of the solvent), then the weight should increase, so using weight loss seems like is not correct.
  5. Language should be thoroughly revised as some of the sentences are confusing and some errors can be found.

Reviewer 2 Report

The manuscript reports the influence of cellulose nanocrystal (CNC) on the properties of a hydroxypropyl guar-based film for intelligent food packaging. The topic is interesting and the conclusions are supported by results. The paper could be of interest to researchers in food industry. Therefore, we suggest to accept with minor modifications given as follows.

Since the use of CNC is one of the major parts of this research, more information on the CNC used should be given. For example, structural parameters of CNC particles (length, etc), the method producing the CNC which is associated with the pH value of the stock solution used, should be given. 
